# TCL-VS: Temporal Contrastive Learning for Self-Supervised Video Summarization

## Abstract

The goal of video summarization is to extract the most important parts from the original video. Most existing methods are based on supervised learning and they have demonstrated superior performance. However, the scarcity of annotated data is a major obstacle in the video summarization task. To reduce the impact of the scarcity, some weakly-supervised and unsupervised methods were proposed. Although they manifested positive results, existing methods ignore the intrinsic association between video clips. To address it, we introduce a new self-supervised learning method called TCL-VS. Our main insight is that a excellent summary requires not only maintaining the original video content but also eliminating redundant information. Inspired by the observation, this work consists of two separate modules that respectively conduct temporal consistency and diversity assessment of video clips. Each module predicts a sequence score by clip, and then we combine them using a weighted method. Extensive experiments demonstrate that our method achieves state-of-the-art performance on two video summarization benchmarks: SumMe and TVSum.

## 1 Introduction

Video summarization has great values in widespread applications, such as video compression, video editing and video retrieval. With the rapid development of video-sharing platforms and the explosive growth of internet video content, the importance of video summarization has become increasingly prominent. Video summarization is the process of extracting meaningful clips/frames from the original video by analyzing the structure and spatiotemporal redundancy of the video in an automatic manner, which attracted increasing attention from both academia and industry.

The early video summarization work mostly used manual heuristic algorithms to obtain certain attributes of frames Chu et al. (2015); Khosla et al. (2013). With the development of deep learning, video summarization tasks have used RNN models Zhang et al. (2016) and attention mechanisms Fajtl et al. (2019); Jung et al. (2020), which advance superior performance.

Most of existed methods employ supervised learning for training. Some general video summary datasets, such as TVSum Song et al. (2015) and SumMe Gygli et al. (2014), provide ground-truth annotations in the form of frame or shot level importance scores. Although supervised approach has achieved excellent results Zhu et al. (2021); He et al. (2023), there are some obstacles to the method, which require extensive resources to construct annotated video summaries. Some weakly-supervised Panda et al. (2017b); Cai et al. (2018); Chen et al. (2019) and unsupervised Mahasseni et al. (2017); Zhou et al. (2018); Zhang et al. (2018) methods have been proposed to address these limitations.

Previous unsupervised video summarization methods Zhou et al. (2018); Zhao et al. (2020) learn the summary by combining the principles of reinforcement learning with hand-crafted reward function for the specific required attributes of the summary. Some existed weakly-supervised video summarization methods Cai et al. (2018); Panda et al. (2017a) use weak label (such as video-level metadata) learning video summarization models that are cheaper than ground-truth data. However, the above methods do not utilize the intrinsic association of the video.

To address this limitation, we first convert the video from fine-grained frames to fine-grained clips by using KTS Potapov et al. (2014), which splits the video into non-overlapping video clips. In-

spired by existed video representation work Dave et al. (2022; 2023), we then introduce temporal contrastive learning into these video clips and two models are trained in this work to capture different aspects of video clips: temporal consistency and diversity. The video clip consistency model aims to identify key clips, while the video clip diversity model focuses on identifying and filtering out redundant clips. The prediction results of these two models are then combined through weighting and summation to derive an importance score for each clip. Finally, the knapsack algorithm, commonly used in previous method Zhu et al. (2021), is employed to generate the final video summary based on the obtained clip scores.

The main contributions of our work are as follows:

- To our best knowledge, this is the first work that applies temporal contrastive self-supervised learning to video summarization. It overcomes the challenge of limited annotated video summaries without the requirement of additional annotations.
- To take full advantage of the intrinsic association between video clips, we devise two models which realize consistency and diversity prediction of video clips.
- To better unit models, we improved the method of weight calculation from the previous work Dave et al. (2023), using all the video clips to calculate the weights of the two models to achieve predictions more efficiently.

Extensive experiments on video summarization benchmark datasets manifest that our self-supervised method significantly outperforms the state-of-the-art unsupervised methods and most supervised methods.

## 2  RELATED WORK

The existing video summarization methods could be cast into two main categories: supervised approaches and unsupervised approaches. This section briefly overviews these categories and contrastive self-supervised learning.

### 2.1  SUPERVISED VIDEO SUMMARIZATION

The recent supervision work has been based on manually annotated datasets. Early deep learning-based methods attempted to estimate the importance of frames by modeling their temporal dependencies. dppLSTM Zhang et al. (2016) used Long short-term memory (LSTM) units to model the variable range time dependence between video frames. H-RNN Zhao et al. (2017) then proposed a two-layer LSTM structure. After the emergence of RNN and transformer, HSA-RNN Zhao et al. (2018) integrated shot segmentation and video summarization into a layered RNN and VASNet Fajtl et al. (2019) applied the attention mechanism to the summarization model. Recently, iPTNet Jiang & Mu (2022) jointly trained video summarization tasks and related moment localization tasks, utilizing additional moment localization data samples to improve the performance of video summarization. A2SummHe et al. (2023), on the other hand, used multimodal enhancement of summarization generation.

### 2.2  UNSUPERVISED VIDEO SUMMARIZATION

The earliest work in the utilization of GANs for learning how to generate a video digest for the accurate reconstruction of the original video was SUM-GAN Mahasseni et al. (2017) in the field of unsupervised deep learning. Afterwards, DR-DSN Zhou et al. (2018) approached video summary generation by formulating it as a sequential decision-making process and designing reward functions to generate diverse and representative video summaries. UnpairedVSN Rochan & Wang (2019) introduced a new method to learning video summaries from unpaired data. Lastly, SUM-GDA Li et al. (2020) implemented globally diverse attention for video summarization.

### 2.3  CONTRASTIVE SELF-SUPERVISED LEARNING

There have been many extensions of contrastive learning in the video domain, following the success of contrastive learning approaches of self-supervised image representation learning such as SimCLR

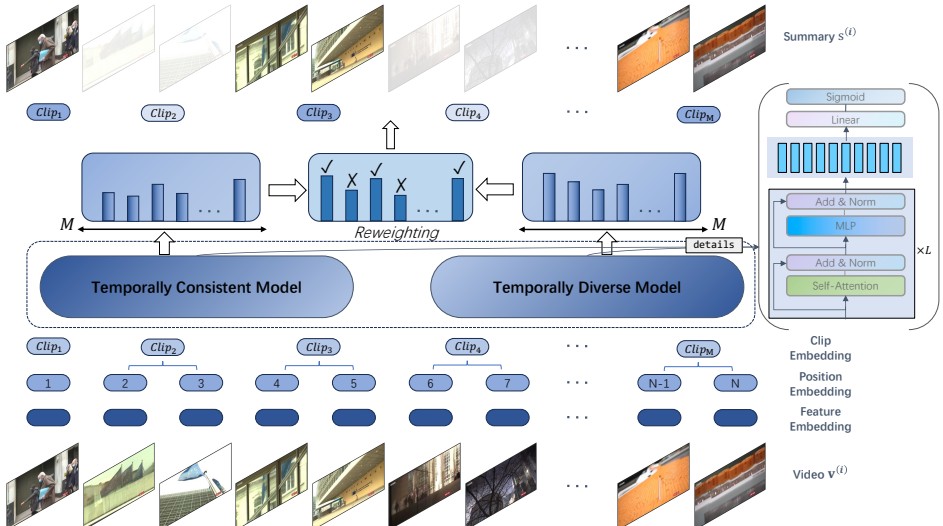

Figure 1: The overview of TCL-VS framework. The input $\{clip_i\}_{i=1}^{M}$ is given to the models to get their predictions. Afterwards, we calculate the predictive weights of the two models and output the important scores of each clip. Finally, to generate a summary, we employ the 0/1 knapsack algorithm to select 15% of the initial video content.

Chen et al. (2020). VTHCL Yang et al. (2020), which employs the SlowFast architecture Feichtenhofer et al. (2019), uses contrastive loss with the slow and fast pathway representations as the positive pair. VIE Zhuang et al. (2020) is proposed as a deep neural embedding-based method to learn video representation in an unsupervised manner.

TCLR Dave et al. (2022) introduces temporal contrastive losses for both temporally pooled and unpooled features to learn the temporal distinctiveness. TimeBalance Dave et al. (2023), following TCLR, proposed a new way to learn temporal consistency. We employed the contrastive loss function of above methods to learn both the temporal consistency model and the temporal diversity model in our approach and the structure of models is like Transformer Encoder.

## 3 PROPOSED APPROACH

To begin with we will provide a brief task definition. Let $V = \{frame_i\}_{i=1}^{N}$ represent an unprocessed video, where $N$ is the number of frames. The method seeks for a set of key frames as a video summary. We represent the set of key frames of the video summary with $S = \{s_j\}_{j=1}^{M}$, where $M$ is the number of frames in the subset $S$. Usually, $M$ is less than a predetermined proportion (such as $15\%$) of $N$. Recent methods have treated the video summarization task as a sequence prediction task, by designing some objective functions to represent the characteristics of superior video summary, and then optimizing these objective functions to obtain more reasonable prediction results. The importance score for each frame is predicted, and each clip (divided by KTS) score is calculated using the mean of the frame scores within the clip. Finally, key clips are selected to form $S$. Our algorithm is the same as the mainstream.

In this section, we elaborate a self-supervised video summarization method as shown in the Figure 1. The overall architecture can be divided into four parts: Input Embedding, Temporal Contrastive Models, Model Reweighting and Fine-Tuning.

### 3.1 INPUT EMBEDDING

Similar to previous work Zhou et al. (2018); Zhu et al. (2021); He et al. (2023), we adopted a universal method to extract features, specifically using pre-trained GoogleNet to extract C-dimensional feature vectors for each frame. In order to better utilize the time correspondence information between video frames, we added a learnable position embedding. Meanwhile, we observed that previ-

ous work mostly inputted the model in frames, while the output was in the form of clips. Therefore, we incorporated clip embedding into the input embedding by applying kernel temporal segmentation (KTS) Potapov et al. (2014). In this way, the video is transformed from N frames to M clips. After adding these embeddings, we denote the generated video features as $V \in R^{M \times C}$.

## 3.2 TEMPORAL CONTRASTIVE MODELS

A promising video summary generation framework should not only extract the most representative fragments but also remove redundant information fragments. Therefore, we used two models to address these two requirements systematically. The first model, the temporal consistent model, focuses on learning common information in videos and assigns a score to each clip, reflecting its representativeness in the video. The second model, the temporal diverse model, is designed to learn the diversity of each clip and assigns a score to measure its diversity in the video. The specific design and training of the models is as follows.

### 3.2.1 NETWORK ARCHITECTURE

Inspired by the excellent performance of transformer Vaswani et al. (2017) in various sequential tasks, we adopt the transformer architecture for our network.

### 3.2.2 TEMPORALLY CONSISTENT MODEL

We adopt the TimeBalance Dave et al. (2023) method as a basis, utilizing distinct clip pairs from the same video as positive samples, and clip pairs from different videos as negative samples. To facilitate the processing of videos, denoted as $V_i = \{x_t^{(i)}\}_{t=1}^n$, we align the number of video clips within a batch. The video clips are represented by the vector projections from the model, recorded as $\{c_t^{(i)} | t \in 1 \cdots n\}$. The contrastive loss function can be expressed by the following equation:

$$L_{con}^{(i)} = - \sum_{\substack{t_1, t_2 \\ t_1 \neq t_2}}^{n} \log \frac{h(c_{t_1}^{(i)}, c_{t_2}^{(i)})}{\sum_{\substack{j=1 \\ j \neq i}}^{B} h(c_{t_1}^{(i)}, c_{t_1}^{(j)}) + h(c_{t_1}^{(i)}, c_{t_2}^{(j)})} \tag{1}$$

This loss is illustrated in Figure 2. The similarity between vectors $c_1$ and $c_2$ is computed using the function $h(c_1, c_2) = exp(c_1^T \cdot c_2 / (\|c_1\| \cdot \|c_2\| \tau))$, which includes an adjustable temperature parameter $\tau$.

### 3.2.3 TEMPORALLY DIVERSE MODEL

We adopt the TCLR Dave et al. (2022) global-local method as a basis, comparing the entire video output result with the video clip output result for loss. Positive sample pairs indicate identical clips, while negative sample pairs indicate different clips. This loss is illustrated in Figure 3 and the contrastive loss function can be expressed by the following equation:

$$L_{div}^{(i)} = - \sum_{t_1=1}^{N} \log \frac{h(c_{t_1}^{(i)}, c_{t_1}^{(i')})}{\sum_{\substack{t_2=1 \\ t_2 \neq t_1}}^{N} h(c_{t_1}^{(i)}, c_{t_2}^{(i')})} \tag{2}$$

Where $N$ is numbers of video clips in $video^{(i)}$ and similarity function $h(c_1, c_2)$ is consistent with the previous text.

## 3.3 MODEL REWEIGHTING

To effectively combine the consistent and diverse information of video clips, an efficient weighting method is required, as depicted in Figure 4. just like TimeBalance Dave et al. (2023) For video instance $v^{(i)}$, we first calculate the cosine similarity for each pair of video clips generated by the model, resulting in two $M \times M$ matrices as $C_C$ and $C_D$. $C_C(c_{t_1}, c_{t_2})$ represent the cosine similarity between the projections of the $c_{t_1}^{(i)}$ and $c_{t_2}^{(i)}$. Afterward, we combine the two matrices and replace

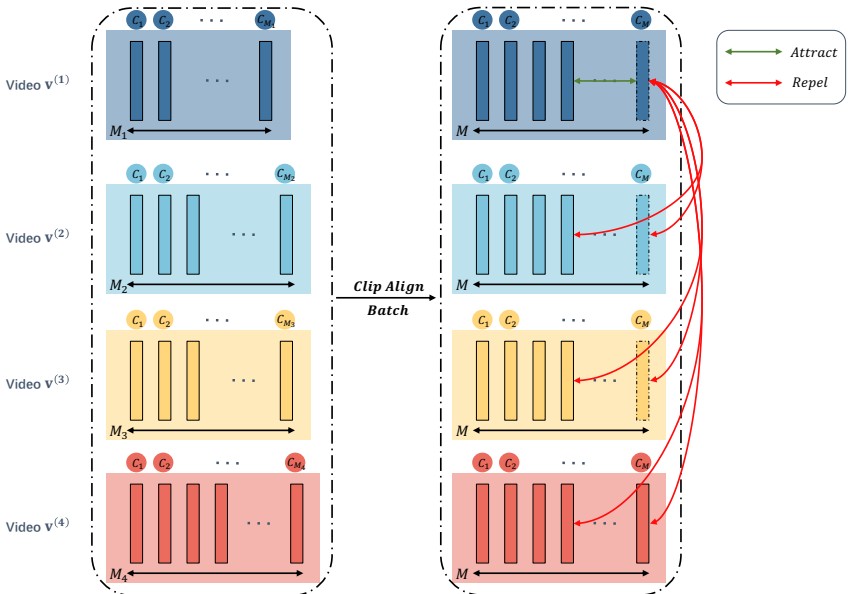

Figure 2: **Temporally Consistent Contrastive Loss** The videos $(v^{(1)}, v^{(2)}, v^{(3)}, v^{(4)})$ are in one batch. First, align the number of clips in the videos by some methods, such as repeating zero clip or the last clip of the video. Then, perform a contrastive loss on each clip of all videos within a batch. Positive samples are adjacent clips, while negative samples are corresponding clips of different videos. As shown in equation 1, the bidirectional green arrows in the figure represent positive sample pairs, while the bidirectional red arrows represent negative sample pairs .

the diagonal with zeros, yielding a new matrix $C$. Next, we follow the formula below to obtain S to derive the value $s^{(i)}$.

$$s^{(i)} = \frac{1}{2M(M-1)} \sum_{\substack{t_1,t_2=1 \\ t_2 \neq t_1}}^{M} (C_C^{(i)} + C_D^{(i)}) \tag{3}$$

Finally, we obtain the final weighted equation as follows:

$$p^{(i)} = s^{(i)} \cdot p_C^{(i)} + (1 - s^{(i)}) \cdot p_D^{(i)} \tag{4}$$

Where $p_C^{(i)}, p_D^{(i)}$ represent the prediction vectors of the consistent and diverse models and $p^{(i)}$ represents the final prediction vector of the framework.

### 3.4 FINE-TUNING

In order to compare fairly with previous SOTA work from ranked-based metrics (Kendall's $\tau$ and Spearman's $\rho$), we fine-tuned the trained model in a supervised manner using ground truth data from standard datasets. We calculated the Kullback-Leibler Divergence between the ground truth scores and the predicted scores as the loss function to optimize the network.

$$D_{KL}(s_{gt}|s) = \sum_{i=1}^{N} s_{gt}^{(i)} \log \frac{s_{gt}^{(i)}}{s^{(i)}} \tag{5}$$

Where $N$ is the number of video frames and $s_{gt}/s$ are ground truth scores and predicted scores.

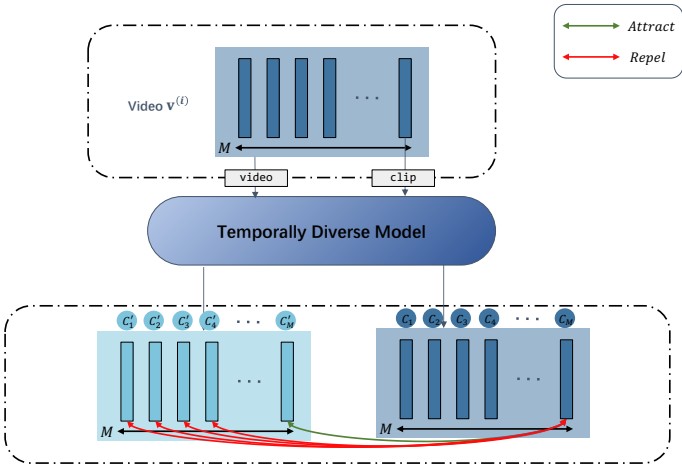

Figure 3: **Temporally Diverse Contrastive Loss** For each video, we input the model from two fine-grained perspectives. The first perspective involves obtaining $\{C_i\}_{i=1}^{M}$ as a whole based on the video instance. The second perspective involves obtaining $\{C_i^{'}\}_{i=1}^{M}$ as a segmented input based on the video clip. We perform a contrastive loss for each clip, as shown in equation 2. Positive sample pairs represent two outputs of the same clip, while negative sample pairs represent two outputs of different clips.

## 4 EXPERIMENTS

### 4.1 EXPERIMENTAL SETUP

The experimental setup section will be divided into three parts. Firstly, we will introduce the dataset used. Then, we will discuss the evaluation metrics used in comparison to other works. Lastly, we will explain some experimental implementation details.

#### 4.1.1 DATASETS.

We train two temporal models on two datasets: QVHighlights Lei et al. (2021) and Breakfast Kuehne et al. (2014). QVHighlights comprises 10,148 videos, each lasting for 150 seconds. It also includes 18,367 moments and 10,310 queries. Breakfast consists of 1712 third-person view videos, with 48 action classes specifically focused on preparing breakfasts. The average length of the videos is two minutes, but there is a significant variation in their duration. On average, each video contains seven instances of actions. For the two training datasets mentioned, we solely relied on the original video data and did not make use of any annotation information.

We evaluate our framework on two benchmarks: SumMe Gygli et al. (2014) and TVSum Song et al. (2015). SumMe comprises 25 user videos, encompassing a wide array of topics like cooking and sports. The duration of these videos ranges from 1 to 6 minutes, and each video is annotated by 15 to 18 individuals, resulting in multiple ground truth summaries for each video. In contrast, TVSum is composed of 50 YouTube videos, covering topics such as news and documentaries. The duration of these videos varies from 2 to 10 minutes.

#### 4.1.2 EVALUATION METRIC.

Following previous work Zhou et al. (2018); Zhu et al. (2021), the F1-score was used to assess the agreement between the generated summaries and the ground-truth summaries. Precision ($P$) and recall ($R$) were computed for both the generated summary ($V_{gs}$) and its corresponding annotated

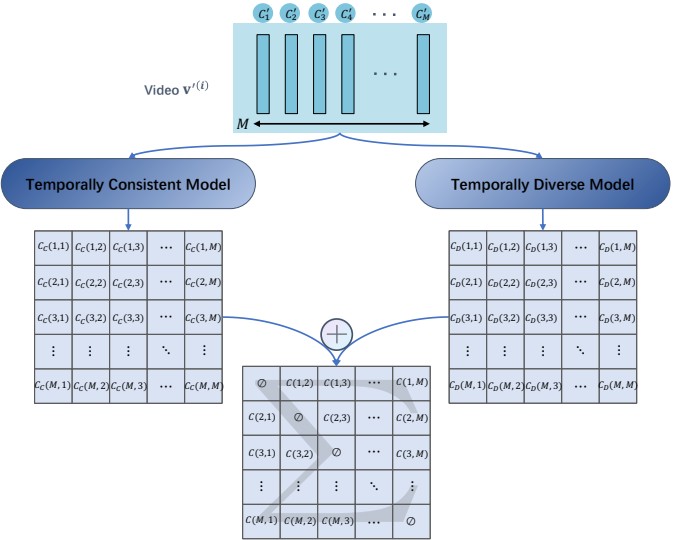

Figure 4: **Temporal Models Reweighting** Firstly, two models were used to obtain $M$ 128-dimensional projections by inputting 1024-dimensional video features $v^{(i)}$. Afterwards, the cosine similarity between the $M$ projections was calculated to obtain two matrices of $M \times M$, with a diagonal value of 1 assigned to 0. Finally, the corresponding position elements of the two matrices were added and summed to obtain an average. As shown in equation 4, the resulting $s^{(i)}$ was used as the weight of Consistent Model, while $1 - s^{(i)}$ was used as the weight of Diverse Model.

summary ($V_{gt}$). The calculation formula is:

$$P = \frac{V_{gs} \cap V_{gt}}{V_{gt}}, R = \frac{V_{gs} \cap V_{gt}}{V_{gs}} \tag{6}$$

The F-score ($F$) was then calculated by:

$$F = \frac{2 \times P \times R}{P + R} \tag{7}$$

In addition, previous work Otani et al. (2019) has indicated that the use of random strategies to generate video summaries may yield a relatively high score. This study also employed the correlation coefficient-based evaluation index proposed in this work to validate our method. The evaluation metrics used were Kendall's $\tau$ and Spearman's $\rho$, which compare the ranking of frames based on their ground truth and predicted scores.

### 4.1.3 IMPLEMENTATION DETAILS.

Following the previous work Zhang et al. (2016); Mahasseni et al. (2017), we extract 1024-dimensional visual features from the pool5 layer of GoogLeNet Szegedy et al. (2015) pre-trained on ImageNet for datasets. To reduce alignment operations in the later training phase, KTS Potapov et al. (2014) is applied to each video in the dataset, and the data is presorted based on the clip count. The number of heads in the multi-head attention layer is set to 8. The hidden size of the video summarization models is set to 128. We set the learning rates of Consistent Model and Diverse Model to 0.0002. We adopt the Adam optimizer to update our models and we maintain weight decay of 1e-5.

The training for QVHighlights and Breakfast lasts 100 epochs, while SumMe or TVSum training lasts for the same number of epochs. In the case of contrastive losses, a default temperature of 0.1 is set.

To ensure fair comparison using ranked-based metrics (Kendall's $\tau$ and Spearman's $\rho$), we employ five different data splits for training and evaluating of standard datasets (SumMe and TVSum) and

utilize the five splits provided in prior work Zhu et al. (2021) along with the fine-tuning method discussed in the previous section.

## 4.2 EVALUATION RESULT.

We compare the proposed method TCL-VS with previous state-of-the-art (SOTA) unsupervised methods on SumMe Gygli et al. (2014) and TVSum Song et al. (2015) datasets in Table 1 and TCL-VS achieves the best results on both datasets. We observe that F1 score in SumMe dataset of our TCL-VS is consistent with CLIP-It Narasimhan et al. (2021), which also utilizes a transformer structure but integrates multimodal information. However, we achieve the same effect by utilizing only a single modality.

Table 1: Overall performance (measured in terms of F-score %) on SumMe and TVSum datasets. The first row presents the results of unsupervised methods, while the second row presents the result of our method.

| Methods | SumMe | TVSum |
|---|---|---|
| SGAN (2017) | 38.7 | 50.8 |
| SUM-FCN (2018) | 41.5 | 52.7 |
| DR-DSN (2018) | 42.1 | 58.1 |
| Sum-Graph (2020) | 49.8 | 59.3 |
| CLIP-It$_{uns}$ (2021) | 50.0 | 59.9 |
| RSGN$_{uns}$ (2021) | 42.3 | 58.0 |
| Ours | **50.2** | **61.9** |

Table 2: The results on SumMe and TVSum, using Kendall's $\tau$ and Spearman's $\rho$, are presented in the following table. The first row contains the results computed using random scores, human scores, and ground truth scores respectively. The methods in the second row are unsupervised methods, while those in the third row are supervised methods. The best results of unsupervised methods and supervised methods are highlighted in **bold** and underline.

| Methods | SumMe | | TVSum | |
|---|---|---|---|---|
| | $\tau$ | $\rho$ | $\tau$ | $\rho$ |
| Random (2019) | 0.000 | 0.000 | 0.000 | 0.000 |
| Human (2019) | 0.205 | 0.213 | 0.177 | 0.204 |
| Ground Truth | 1.000 | 1.000 | 0.364 | 0.456 |
| SGAN (2017) | - | - | 0.024 | 0.032 |
| DR-DSN (2018) | 0.047 | 0.048 | 0.020 | 0.026 |
| RSGN$_{uns}$ (2021) | 0.071 | 0.073 | 0.048 | 0.052 |
| DSNet (2021) | 0.051 | 0.059 | 0.108 | 0.129 |
| RSGN (2021) | 0.083 | 0.085 | 0.083 | 0.090 |
| CLIP-It (2021) | - | - | 0.108 | 0.147 |
| Sum-Graph (2020) | - | - | 0.094 | 0.138 |
| iPTNet (2022) | 0.101 | 0.119 | 0.134 | 0.163 |
| A2Summ (2023) | 0.108 | 0.129 | 0.137 | 0.165 |
| TCL-VS | **0.076** | **0.103** | **0.054** | **0.079** |
| TCL-VS + fine-tuning | 0.097 | 0.132 | 0.118 | 0.171 |

## 4.3 ABLATION STUDIES.

### 4.3.1 CONTRIBUTION OF DIFFERENT COMPONENTS

In this study, as shown in Table 3, the impact of each model ($M_I$, $M_D$) and reweighting scheme on the SumMe and TVSum dataset is analyzed. Initially, the performance of the $M_I$ and $M_D$ models individually is demonstrated (Row 1-2). Subsequently, the framework performs optimally when the predictions of both models are averaged (Row 3). Finally, the effectiveness of the reweighting scheme using temporal similarity is illustrated (Row 4). We can observe that the weighted mean

method (Row 3) is slightly superior in terms of rank correlation coefficient but shows a significant difference in F-score.

Table 3: Ablation study of different components of our framework in standard datasets. '$M_C$','$M_D$','$Re$' represent temporally consistent model, temporally diverse modal and model reweighting.

| ID | $M_C$ | $M_D$ | $Re$ | SumMe | | | TVSum | | |
|----|-------|-------|------|-------|-------|-------|-------|-------|-------|
| | | | | F1 | $\tau$ | $\rho$ | F1 | $\tau$ | $\rho$ |
| 1 | ✓ | ✗ | ✗ | 48.2 | 0.030 | 0.039 | 58.0 | 0.035 | 0.050 |
| 2 | ✗ | ✓ | ✗ | 41.8 | 0.060 | 0.081 | 59.3 | 0.056 | 0.082 |
| 3 | ✓ | ✓ | ✗ | 46.3 | **0.080** | **0.108** | 61.2 | **0.058** | **0.084** |
| 4 | ✓ | ✓ | ✓ | **50.2** | 0.076 | 0.103 | **61.9** | 0.054 | 0.079 |

### 4.3.2 DIFFERENT CLIP ALIGNMENT AND TRAINING METHODS

As shown in Figure 2, when training the Consistent model, it is necessary to align the clips first. We have adopted three alignment methods for videos within a batch. The first method is to fill the end of videos with a small number of clips with the clip projection of all zeros. This ensures that the number of each video clip is consistent with the maximum number in the batch. The second method involves repeatedly filling in the projection of the last video clip at the end. The third method is to mirror and repeat the video clips in reverse until the number of clips is consistent. We have abbreviated these three strategies as zero, copy, and reflection. During specific training, we applied these three strategies to both the large video datasets (QVHighlights and Breakfast) and the standard datasets (SumMe and TVSum). Additionally, we trained 100 epochs on a larger dataset to learn two video clip processing abilities before conducting the training and validation on a standard dataset. Ablation studies were also performed on the training methods, and the detailed results can be found in Table 4.

Table 4: The effect of different clip alignment and training methods on the dataset. 'Pretrain' indicates training only on the QVHighlights and Breakfast datasets, while 'Standard' indicates training only on the SumMe dataset. 'zero', 'copy', and 'reflection' represent the three clip alignment methods.

| Train Data | Clip Fusion | F1 | $\tau$ | $\rho$ |
|------------|-------------|------|--------|--------|
| | zero | 46.6 | 0.036 | 0.048 |
| Pretrain | copy | 49.1 | 0.032 | 0.044 |
| | reflection | 45.9 | 0.031 | 0.044 |
| | zero | 49.0 | 0.072 | 0.095 |
| Standard | copy | 48.7 | 0.061 | 0.099 |
| | reflection | 47.6 | 0.063 | 0.079 |
| | zero + zero | 49.3 | 0.053 | 0.072 |
| | zero + copy | 49.6 | 0.058 | 0.077 |
| | zero + reflection | 49.6 | 0.046 | 0.062 |
| | copy + zero | 49.7 | 0.072 | 0.099 |
| Pretrain + Standard | copy + copy | 48.1 | 0.054 | 0.074 |
| | copy + reflection | **50.2** | **0.076** | **0.103** |
| | reflection + zero | 49.2 | 0.060 | 0.083 |
| | reflection + copy | 48.4 | 0.055 | 0.074 |
| | reflection + reflection | 49.0 | 0.043 | 0.059 |

## 5 CONCLUSION

In this work, we address a new self-supervised video summarization method by training a video summarization framework using two types of temporal contrastive losses. Through extensive experiments on two benchmark datasets, we have demonstrated that our method outperforms other state-of-the-art unsupervised alternatives. Moreover, the results obtained are comparable to or better than most supervised methods.

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
