# OpenReview forum: "TCL-VS: Temporal Contrastive Learning for Self-Supervised Video Summarization"
_ICLR.cc/2024/Conference — ICLR 2024 Conference Withdrawn Submission_

### Official Review · Reviewer_Jbri · 2023-10-27

**Soundness:** 1 poor
**Presentation:** 2 fair
**Contribution:** 1 poor
**Rating:** 3
**Confidence:** 4

**Summary:**

The paper proposes TCL-VS. which is a video summarization model that utilizes Temporal Contrastive Learning (TCL). Due to the lack of labeled data, the author chose to use an unsupervised learning method to address this problem. The author claims that a good video summary should extract most representative segments from the video while removing redundant segments with similar information. To achieve these two goals, the author uses two models, Temporal Consistency model, which measures the representativeness of clips in the video, and Temporal Diversity model, which is designed to measure the diversity of clips in the video. In order to combine information of the two models, they used a model reweighting method. Lastly, to compare with prior SOTA works, the author uses KL-divergence when fine-tuning on the two video summarization datasets, SumMe and TVSum.

**Strengths:**

The paper is the first work which applies Temporal Contrastive Learning (TCL) to utilize the intrinsic association of the video. They achieve a good performance compared to the other methods and show that their methods are effective in ablation studies.

**Weaknesses:**

The most significant weakness of this paper comes from the lack of novelty. While it is true that this is the first study to use TCL, the paper tends to rely heavily on ideas taken from other prior research without much thought or consideration. The methods used in section 3.2 are all from other papers such as “Attention is all you need (2017)”, “TimeBalance, Dave et al. (2023)”, “TCLR, Dave et al. (2022)”, leading to serious lack of novelty. The paper would have been more interesting if the author presented an analysis of why attracting similar clips and repelling clips from other videos can extract most representative segments, even though different videos may have similar categories or structures.

Another weakness is the experimental setting. Due to data scarcity, even though the author uses five different data splits for training and evaluating, the F1 score can be noisy because small changes in the dataset can lead to large changes in the metric. In other words, depending on what training and evaluation data was sampled, can highly vary the performance result. The paper might be more persuasive if the performance was reported with the mean and standard deviation, after running the TCL-VS model at least five times.

**Questions:**

1. Suggestion: Explanation of $c_{t_1}$, and $c_{t_2}$ was not written, and instead, was written in TimeBalance, Dave et al. (2023). This made the reviewer hard to understand when reading the paper. (Lack of clarity) Please write a more detailed explanation of the equation.

2. As stated in the weakness section, for the paper to be stronger, the author should provide TCL-VS model performance results with the mean and standard deviation after running at least five times with different random seeds.

3. We wonder if the authors have set-aside some videos for validation purpose. If so, how many videos are used for validation? If not, how the authors have chosen a model?

---

### Official Review · Reviewer_PYMF · 2023-11-01

**Soundness:** 2 fair
**Presentation:** 2 fair
**Contribution:** 2 fair
**Rating:** 5
**Confidence:** 3

**Summary:**

The authors introduce temporal contrastive learning into video summarization to utilize the intrinsic association of the videos. They trained two models to capture temporal consistency and temporal diversity respectively, and then combine the results of the two models by weighting and summation to get the clip-level scores. The video clip consistency model is employed with temporally consistent contrastive loss to identify key clips, and the video clip diversity model is employed with temporal diversity contrastive loss to filter out redundant clips.

**Strengths:**

- This paper introduce temporal contrastive self-supervised learning to video summarization, which is useful compared with unsupervised and supervised baselines.

- Several experiments are carried out on SumMe and TVSum datasets with detailed ablation study.

**Weaknesses:**

- The motivation for this paper has not been very elaborately explained, and it appears somewhat vague.

- This paper primarily utilizes a combination of existing methods with very minor improvements, lacking novelty.

**Questions:**

- Is there any examples you can provide to explain why the previous methods do not utilize the intrinsic association of the videos?

- Does the parameters double in this method, since it utilizes two models? Would if be fair compared with the baseline?

- This paper does show effectiveness on the existing datasets, although it looks unpolished and hastily completed. I will make my final decision based on other reviewers' comments and the authors' rebuttal.

---

### Official Review · Reviewer_4CFj · 2023-11-02

**Soundness:** 1 poor
**Presentation:** 2 fair
**Contribution:** 2 fair
**Rating:** 3
**Confidence:** 4

**Summary:**

- This work introduces TCL-VS, a self-supervised video summarization method inspired by the TCLR and TimeBalance papers.
- TCL-VS consists of two separate modules that respectively conduct temporal consistency and diversity assessment of video clips. Each module predicts a sequence score by clip, and then combine them using a weighted method.
- The proposed framework is evaluated on two benchmark datasets: TVSum and SumMe.

**Strengths:**

Video summarisation is a challenging area to work on and I like the overall idea of using time-variant (temporally diverse) or time-invariant (temporally consistent) representations for summarisation.

**Weaknesses:**

- The explanation and writing can be improved; it is very hard to follow. Could you please share the code with me so I can understand the method better? Additionally, I suggest adding pseudocode in the updated version.
- Could you please update Tables 1 and 2 with additional details, such as model architecture, parameters, and input setup? This will allow us to make a more fair and informed comparison with other models.
- How did you utilize the "1024-dimensional visual features from the pool5 layer of GoogLeNet"?
- Did you keep TCLR and TimeBalance frozen, or did you train them from scratch with your data (TVSum and SumMe)? If you tuned them, did you use the weights from TCLR and TimeBalance?
- I am missing the details of the hyperparameters and other training setup.

**Questions:**

Please see weakness.

While I appreciate the high-level idea of this paper, it seems incomplete in its current form. I suggest that the authors rework on it for a future submission.